# From Stress to Synapse: The Neuronal Atrophy Pathway to Mood Dysregulation

**DOI:** 10.3390/ijms26073219

**Published:** 2025-03-30

**Authors:** Henry Krasner, Claire Victoria Ong, Paige Hewitt, Thomas A. Vida

**Affiliations:** Kirk Kerkorian School of Medicine, University of Nevada, Las Vegas, 625 Shadow Lane, Las Vegas, NV 89106, USA; krasnh1@unlv.nevada.edu (H.K.); ongc5@unlv.nevada.edu (C.V.O.); hewitp1@unlv.nevada.edu (P.H.)

**Keywords:** neuronal atrophy, mood disorders, neuroinflammation, synaptic plasticity, brain-derived neurotrophic factor (BDNF), glucocorticoids, mitochondrial dysfunction, neurotrophic hypothesis, depression pathophysiology, longitudinal neuroimaging

## Abstract

Mood disorders, including major depressive disorder and bipolar disorder, are among the most prevalent mental health conditions globally, yet their underlying mechanisms remain incompletely understood. This review critically examines the neuronal atrophy hypothesis, which posits that chronic stress and associated neurobiological changes lead to structural and functional deficits in critical brain regions, contributing to mood disorder pathogenesis. Key mechanisms explored include dysregulation of neurotrophic factors such as brain-derived neurotrophic factor (BDNF), elevated glucocorticoids from stress responses, neuroinflammation mediated by cytokines, and mitochondrial dysfunction disrupting neuronal energy metabolism. These processes collectively impair synaptic plasticity, exacerbate structural atrophy, and perpetuate mood dysregulation. Emerging evidence from neuroimaging, genetic, and epigenetic studies underscores the complexity of these interactions and highlights the role of environmental factors such as early-life stress and urbanization. Furthermore, therapeutic strategies targeting neuroplasticity, including novel pharmacological agents, lifestyle interventions, and anti-inflammatory treatments, are discussed as promising avenues for improving patient outcomes. Advancing our understanding of the neuronal atrophy hypothesis could lead to more effective, sustainable interventions for managing mood disorders and mitigating their global health burden.

## 1. Introduction

### 1.1. Prevalence and Impact of Mood Disorders

The diagnosis of a mood disorder encompasses a spectrum of mental health conditions featuring different disturbances in emotional regulation [1,2]. The two most notable mood disorders are depression and BD [3,4]. Persistent feelings of sadness and a loss of interest in previously enjoyed activities characterize conditions such as major depressive disorder (MDD) [5] and persistent depressive disorder (PDD) [6]. Bipolar disorder features periods of elevated mood or mania and potentially episodes of depression [7,8]. During these depressive episodes, patients may experience changes in sleep, appetite, energy, and concentration [9,10,11,12]. In manic episodes, heightened energy, impulsivity, racing thoughts, and feelings of grandiosity occur [2,13,14,15,16]. These disorders significantly impact daily functioning and quality of life. Current treatment options have varying degrees of success for patients and may induce additional problematic symptoms [17,18,19]. Identifying the exact pathophysiology behind these clinical manifestations may yield more effective treatment options in the future.

Mood disorders encompass many of the most common mental disorders globally, including depression and bipolar disorder (BD) [20,21,22]. The prevalence of mood disorders has increased over the past four decades. A systematic review and meta-analysis indicated that the global prevalence of common mental disorders, including mood disorders, remained high, with substantial regional variations, covering the period from 1980 to 2013 [23]. Additionally, a longitudinal study in the UK reported a significant increase in adolescent conduct and emotional problems from 1974 to 1999 [24]. In Denmark, the incidence rates of diagnosed BD and depression increased significantly from 1995 to 2010, particularly among individuals up to 29 years of age [25]. In Germany, in-patient admissions for BD among individuals aged up to 19 increased by 68.5%, from 1.13 to 1.91 admissions per 100,000 people, between 2000 and 2007 [26]. From 2005 to 2017, major depressive episodes rose by 52% among adolescents aged 12 to 17 and by 63% among young adults aged 18 to 25, with severe psychological distress and suicide-related outcomes also surging significantly in these age groups [27].

In the United States, more than one in ten individuals aged 17–39 years may have a mood disorder according to previous studies, and this value is even higher in vulnerable populations, such as older patients and low socioeconomic status (SES) communities [20,28,29]. Approximately 8 million individuals worldwide die per year due to the impact of mental disorders, including mood disorders. This morbidity amounts to 14.3% of total deaths worldwide annually [30]. Mood disorders often accompany other psychiatric conditions and lead individuals to participate in activities that may place them at risk of adverse consequences [31,32,33]. While multiple treatment options are available for mood disorder patients, relapse following treatment is standard, with some mood disorder subtypes being notoriously treatment-resistant and resulting in worse patient outcomes [19,34,35]. Some mood disorders, such as depression, are so prevalent that they are considered a significant contributor to the global burden of disease, according to the World Health Organization [1]. Understanding physiological mechanisms underlying mood disorders may allow for the development of improved and novel solutions to combat them and decrease this significant public health crisis.

### 1.2. Reevaluating the Neurotransmitter Dysregulation Hypothesis and Its Limitations

The serotonin hypothesis of depression, which has long been a cornerstone of understanding mood disorders, has increasingly come under scrutiny. Recent systematic reviews, such as Moncrieff et al., show no consistent evidence directly linking reduced serotonin levels or activity to depression [36]. Emerging critiques that support this view argue that the original serotonin hypothesis oversimplifies the neurobiology of depression [37,38]. While serotonin’s involvement in mood regulation remains acknowledged, its role appears more complex and may pertain to a specific subset of individuals with depression. For example, serotonin may still be relevant in modulating emotional and cognitive processes in patients with heightened emotional dysregulation [38,39,40]. However, broader perspectives increasingly emphasize the need to consider multiple neurotransmitters and systemic factors, such as dopamine, norepinephrine, and inflammation, which play significant roles in the pathophysiology of depression [37,41]. Serotonin’s involvement in reward learning, emotional regulation, and the default mode network all contribute to the broader understanding of mood disorders [41,42,43].

Beyond serotonin, growing evidence suggests that dysregulation in excitatory and inhibitory neurotransmission—particularly the glutamatergic and GABAergic pathways—plays a central role in the pathophysiology of mood disorders [44]. The glutamatergic system, which governs synaptic plasticity and cognitive function, is often hyperactive in depression, leading to excessive excitatory signaling and increased susceptibility to neurotoxic damage [45]. Chronic stress has been shown to increase extracellular glutamate levels, resulting in prolonged N-methyl-D-aspartate (NMDA) receptor activation, intracellular calcium overload, and oxidative stress, all contributing to neuronal atrophy [46]. Conversely, deficits in GABAergic inhibition weaken the brain’s ability to regulate excitatory activity, leading to heightened neural excitability, disrupted emotional processing, and increased vulnerability to synaptic loss in key brain regions such as the prefrontal cortex and hippocampus [47].

The serotonin reuptake mechanism emerged as a potential molecular target in treating depressive disorders in the 1960s [48,49,50] (see Table 1). Serotonin (5-HT) primarily acts in extracellular spaces, using volume transmission to target postsynaptic serotonin receptors selectively [51]. The development of imipramine and, later, selective serotonin reuptake inhibitors (SSRIs) marked the advent of serotonin-based treatments for major depressive disorder [50]. However, despite their success, these treatments are limited by their inability to fully address the complexities of depression, such as treatment-resistant depression (TRD).

Recent findings suggest that interventions targeting glutamate and GABA signaling may offer novel therapeutic approaches for depression, particularly in cases of TRD [52]. Pharmacological agents such as ketamine, an NMDA receptor antagonist, have been shown to exert rapid antidepressant effects by modulating glutamatergic transmission, promoting synaptic plasticity, and restoring prefrontal connectivity [53]. Similarly, positive allosteric modulators of GABA-A receptors, including neurosteroids, have demonstrated antidepressant potential by enhancing inhibitory tone and stabilizing neural network function [54]. These findings highlight the limitations of serotonin-based treatments and emphasize the importance of a more integrative approach that accounts for neurotransmitter interactions and neuroplasticity mechanisms in mood disorders [55].

### 1.3. Treatment-Resistant Depression: New Approaches and Neuroplasticity

Treatment-resistant depression (TRD) affects approximately 30% of individuals with depression, underscoring the limitations of serotonin-based treatments like SSRIs [56,57]. TRD may not result from serotonin deficiency alone but involves complex interactions between multiple neurotransmitter systems, genetic variations, and neuroplasticity mechanisms [58,59]. Neuroimaging studies reveal structural changes in brain networks involved in cognitive and emotional processing, particularly in the prefrontal cortex and hippocampus [60,61]. These alterations are associated with the persistence of depressive symptoms and highlight the need for treatments targeting the brain’s structural health.

Increasing evidence suggests that TRD may be linked to excitatory and inhibitory neurotransmission disruptions, particularly glutamatergic overactivity and GABAergic deficits [52]. Patients with TRD often exhibit elevated glutamate concentrations in the anterior cingulate cortex and prefrontal cortex, reflecting a state of hyperexcitability that contributes to persistent mood instability, cognitive dysfunction, and impaired treatment response [53]. Excessive NMDA receptor activation in TRD leads to sustained intracellular calcium accumulation, increased oxidative stress, and mitochondrial dysfunction, all exacerbating synaptic loss and preventing neural recovery [62]. Furthermore, reductions in GABAergic function weaken inhibitory control over excitatory networks, heightening vulnerability to stress-induced neuronal damage. The interplay between these excitatory–inhibitory imbalances and neuroplasticity deficits underscores the need for novel, mechanism-based interventions in TRD [63].

Recent advances in neuromodulation techniques, such as transcranial magnetic stimulation (TMS) and deep brain stimulation (DBS), show promise in modulating the neural circuits involved in mood regulation, offering new hope for TRD patients [64]. Novel pharmacological treatments targeting the glutamatergic system, such as ketamine, have also been effective in addressing TRD, providing rapid and sustained antidepressant effects by enhancing neuroplasticity [65,66].

Ketamine’s antidepressant properties stem from its ability to inhibit NMDA receptors, thereby reducing glutamate excitotoxicity while promoting the release of brain-derived neurotrophic factor (BDNF) [67]. This process facilitates synaptogenesis and restores structural connectivity in prefrontal circuits affected by TRD [68]. Additionally, positive allosteric modulators of GABA-A receptors, such as neurosteroids, have emerged as promising therapies for restoring inhibitory tone and preventing hyperexcitability-driven atrophy [69].

Classic psychedelics such as LSD, DMT, and psilocybin also show promise in treating both TRD and other mood disorders, as their agonism of the 5-HT_2A_ receptor primarily affects increased neuroplasticity, including synaptogenesis [70], neurogenesis [71], neurogenesis [72,73,74,75] and reconfiguration of brain network connectivity [76,77,78,79]. Psilocybin may be as effective as SSRIs in treating depression and anxiety [80]. The first randomized trial comparing psilocybin with escitalopram demonstrates secondary outcomes generally favored psilocybin, although with no significant difference in antidepressant effects [81]. Notably, psychedelics indirectly modulate glutamate signaling via serotonergic pathways, increasing AMPA receptor activation and promoting synaptic resilience [82]. This suggests that their antidepressant effects may partly stem from restoring the excitatory–inhibitory balance in brain circuits affected by chronic stress and neuronal atrophy [83].

Most recently, a longer-term follow-up suggested that psilocybin showed more substantial benefits in social functioning, psychological connectedness, and meaning in life, indicating its potential for broader mental health recovery beyond symptom reduction [84]. Yet, excessive or dysregulated serotonin system activation has the potential to contribute to maladaptive neuroplasticity—which may lead to cognitive rigidity, affect instability, or exacerbation of psychiatric symptoms [85]. The limitations of serotonin-based treatments have led to the exploration of other mechanisms, such as neuroplasticity and the role of stress-induced neuronal atrophy [59,65]. The findings from this study provide empirical support for the neuroplasticity hypothesis as a more comprehensive framework for understanding and treating mood disorders, particularly in cases of TRD, where serotonin-based approaches often fail to yield sustained improvements.

### 1.4. Emergence of the Neuronal Atrophy Hypothesis

Neuronal atrophy may be defined as the progressive degeneration and loss of neurons—leading to diminished cell size and functionality. Recent research into the neurobiology of depression has shifted focus from neurotransmitter dysregulation to broader frameworks, such as the neuronal atrophy hypothesis [86,87]. This notion, also known as the neurotrophic hypothesis, suggests that chronic stress leads to structural changes in the brain regions critical for emotion processing and regulation [88,89]. Prolonged exposure to stress hormones during depressive episodes can deteriorate brain structures like the hippocampus and prefrontal cortex [90]. This structural damage contributes to the loss of neuroplasticity, making it harder for individuals to adapt to new stressors and resulting in emotional dysregulation [90,91,92].

One of the primary mechanisms driving neuronal atrophy in mood disorders is excitotoxicity, a process initiated by excessive glutamatergic signaling [44]. Chronic stress elevates extracellular glutamate levels, leading to prolonged activation of N-methyl-D-aspartate (NMDA) receptors [93]. This results in sustained calcium influx into neurons, overwhelming mitochondrial buffering capacity and triggering mitochondrial dysfunction. The accumulation of reactive oxygen species (ROS) further exacerbates neuronal damage by promoting lipid peroxidation, protein misfolding, and apoptosis [94]. These mitochondrial deficits ultimately compromise synaptic integrity, reducing dendritic complexity and contributing to the cortical thinning observed in major depressive disorder (MDD) [95].

In support of this hypothesis, post-mortem analyses show the reduced size of pyramidal neurons in the dorsolateral prefrontal cortex and fewer synapses in patients with major depressive disorder [96,97]. Moreover, neuroimaging studies using magnetic resonance spectroscopy (MRS) have demonstrated lower levels of gamma-aminobutyric acid (GABA) and higher glutamate-to-GABA ratios in patients with depression, reinforcing the idea that excitatory–inhibitory imbalances drive neurodegeneration. GABAergic deficits, in particular, reduce the brain’s ability to regulate excitatory activity, leaving neurons more vulnerable to stress-induced excitotoxicity and atrophy [98]. These findings suggest that neuronal atrophy, rather than neurotransmitter dysregulation alone, may underlie the structural changes observed in depression and mood disorders.

In addition to excitotoxicity and mitochondrial dysfunction, neuroinflammation plays a significant role in stress-induced neuronal atrophy [99]. Chronic stress activates microglia, the brain’s resident immune cells, producing excessive pro-inflammatory cytokines such as interleukin-6 (IL-6) and tumor necrosis factor-alpha [100]. These cytokines contribute to synaptic pruning, impair neurogenesis, and exacerbate oxidative stress, further amplifying the neurodegenerative effects of chronic mood disorders [101]. Elevated C-reactive protein (CRP) levels and inflammatory markers in MDD patients correlate with reduced cortical volume, suggesting systemic inflammation accelerates brain atrophy in susceptible individuals [99].

Moreover, emerging treatments with ketamine and classic psychedelics drugs (discussed previously) modulate neural circuits involved in mood regulation, ultimately resulting in neuroplasticity. Ketamine’s rapid antidepressant effects stem from its ability to inhibit NMDA receptors, thereby reducing glutamate excitotoxicity while increasing BDNF expression, facilitating synaptic recovery [102]. Similarly, psychedelics indirectly influence glutamate signaling through 5-HT2A receptor agonism, promoting synaptogenesis and restoring atrophied neural networks in the prefrontal cortex [73,103].

While the neurotransmitter dysregulation hypothesis, particularly serotonin deficiency, has shaped treatments for mood disorders, it fails to account for the full complexity of these conditions. The neuronal atrophy hypothesis provides a more comprehensive framework, emphasizing structural and plasticity changes in the brain resulting from chronic stress. Both hypotheses acknowledge the importance of serotonin, but the dysregulation hypothesis focuses on synaptic transmission and reuptake. In contrast, the atrophy hypothesis highlights the long-term consequences of serotonin imbalance and stress on brain regions crucial for emotion regulation [91,92].

Unlike the neurotransmitter dysregulation hypothesis, which primarily examines chemical imbalances, the neuronal atrophy hypothesis explains how chronic stress disrupts neurocircuitry through excitotoxicity, mitochondrial dysfunction, and neuroinflammation [99]. Stress-induced increases in glutamate levels lead to excessive NMDA receptor activation, causing intracellular calcium overload, mitochondrial damage, and oxidative stress [104]. These disruptions weaken synaptic connectivity and impair neuroplasticity, ultimately reducing the structural integrity of the prefrontal cortex and hippocampus [105]. At the same time, GABAergic deficits fail to counterbalance excessive excitatory signaling, further increasing neuronal vulnerability to atrophy [45]. Chronic stress also activates microglia, triggering inflammatory cascades that degrade synaptic networks and inhibit neurogenesis [100]. Elevated levels of inflammatory cytokines, such as IL-6 and TNF-α, correlate with reduced cortical volume and worsening depressive symptoms, underscoring the role of neuroimmune dysfunction in stress-related neuronal atrophy [99].

Effective treatment may thus require addressing neurotransmitter balance, neuronal health, and plasticity (see Table 1). Ketamine, an NMDA receptor antagonist, enhances BDNF expression, restores excitatory–inhibitory balance, and promotes synaptic recovery to counteract glutamate excitotoxicity. Anti-inflammatory therapies lower microglial activation and prevent stress-induced synaptic degradation, offering new strategies to protect against neuronal atrophy in treatment-resistant depression. The neuronal atrophy hypothesis explains how disruptions in glutamatergic regulation, mitochondrial function, and neuroimmune interactions drive mood disorders, providing a framework for developing biologically targeted treatments.

### 1.5. Exploring Neuronal Atrophy in Mood Disorders

Although significant research has investigated the various models attempting to explain the molecular mechanisms of mood disorders, an extensive knowledge gap still exists. This critical review aims to evaluate the proposed molecular mechanisms underpinning neuronal atrophy, particularly its role in mood dysregulation. We hypothesize that reduced dendritic complexity and synaptic loss in the prefrontal cortex and hippocampus due to chronic stress-induced neuronal atrophy contribute significantly to the pathogenesis of mood disorders. We aim to enhance the understanding of these mechanisms and support the development of targeted pharmacological treatments that address neurotransmitter dysregulation and structural brain changes.

## 2. Neuroanatomical Alterations in Mood Disorders: Structural and Functional Perspectives

### 2.1. Historical Perspectives on Structural Brain Changes in Mood Disorders

The historical perspective on the neuronal atrophy hypothesis sheds light on understanding structural changes in the brain over time. The basis of this hypothesis is that structural deterioration occurs in the brains of mood disorder patients, specifically in regions responsible for regulating emotion and mood control [106,107,108]. This hypothesis originates from post-mortem examinations that revealed visible structural alterations in the brains of individuals known to have mood disorders, among other psychological disorders, such as schizophrenia. Early researchers identified reductions in neuronal size and density, as well as glial cell abnormalities in recurrent brain regions of these individuals, suggesting a possible connection between physical brain morphology and mood dysregulation [92,109,110,111,112].

### 2.2. Neuroimaging and Post-Mortem Evidence of Neuronal Atrophy

Contemporary neuroimaging techniques, including magnetic resonance imaging (MRI) and positron emission tomography (PET), have allowed for advancement in the investigation of brain structures and functions in living individuals with mood disorders [113,114]. Alterations consistently occur in brain regions implicated in mood regulation, including the prefrontal cortex, hippocampus, and amygdala [115,116] (Table 2). Furthermore, modern post-mortem analyses have provided complementary evidence, revealing cellular and molecular changes in these brain regions [92,106,109,110,111,112]. Reductions in neuronal size, synaptic density, and alterations in neuroplasticity-related molecules occur, emphasizing the role of neuronal atrophy in mood disorders.

### 2.3. Thalamus and Mood Disorders

The thalamus is critical in sensory processing and emotional regulation, acting as a relay hub between subcortical and cortical structures. In major depressive disorder (MDD) and bipolar disorder (BD), structural and functional MRI studies have demonstrated altered thalamo-cortical connectivity and reduced thalamic volume, which are associated with cognitive and affective dysfunction [117]. Additionally, disruptions in thalamic functional connectivity with limbic and striatal circuits contribute to emotional instability and sensory dysregulation [118]. Thalamic dysfunction also differentiates MDD from BD, with MDD patients displaying hyperconnectivity between the thalamus and parietal cortex, while BD patients exhibit disruptions in fronto-thalamic pathways [119]. Furthermore, structural abnormalities in the thalamus have been observed even in drug-naïve, first-episode MDD patients, highlighting the thalamus’ early involvement in mood disorders [120,121].

### 2.4. Basal Ganglia and Reward Processing in Depression

The basal ganglia, particularly the striatum (caudate nucleus, putamen, and nucleus accumbens), play a central role in reward processing, motivation, and mood regulation. Structural and functional imaging studies have demonstrated reduced basal ganglia volume and impaired connectivity in individuals with MDD and BD, contributing to anhedonia (loss of pleasure), psychomotor retardation, and mood instability [122,123]. Genetic analyses further support the basal ganglia’s role in mood disorders, with genome-wide association studies (GWASs) identifying 72 genetic loci associated with basal ganglia structure, many of which overlap with psychiatric disorders, including schizophrenia, bipolar disorder, and depression [124]. Functional MRI findings reveal disruptions in cortico-striatal connectivity in BD, which may underlie abnormal reward sensitivity and impulsive decision-making [118,125]. Additionally, basal ganglia–thalamic circuit dysfunctions have been implicated in mood instability, cognitive control deficits, and impulsivity, further reinforcing their role in affective disorders [124,126].

### 2.5. Orbitofrontal Cortex and Impulse Control Dysfunctions

The orbitofrontal cortex (OFC) is essential for decision-making, impulse control, and affect regulation. Structural MRI studies consistently show reduced OFC volume in individuals with major depressive disorder (MDD), bipolar disorder (BD), and obsessive compulsive disorder (OCD), which correlates with maladaptive emotional responses, risk-taking behavior, and impaired social cognition [127,128]. The OFC plays a critical role in emotional processing and reinforcement learning, and its dysfunction has been linked to rumination, compulsive behaviors, and difficulty adjusting to changing environmental demands [129].

In OCD, hyperactivity in the OFC is believed to reinforce obsessive thought patterns and compulsions, further exacerbating dysfunction in cognitive flexibility and affect regulation [129,130]. Additionally, the OFC’s role in reward processing and inhibitory control suggests that its dysfunction in mood disorders may contribute to behavioral inflexibility and heightened sensitivity to negative feedback, a common feature of MDD and BD [131,132]. These findings highlight the importance of fronto-limbic dysregulation in affective disorders and suggest that the OFC should be a key target for neuropsychiatric interventions.

### 2.6. Cerebellum and Mood Regulation

While traditionally linked to motor control, the cerebellum is increasingly recognized as a key structure involved in cognitive and emotional processing. Neuroimaging studies indicate reduced cerebellar volume in individuals with MDD and BD, which is associated with executive function impairments, attention deficits, and emotional dysregulation [133,134]. The cerebellum strongly connects with the prefrontal cortex and limbic structures, influencing affective control and cognitive processing [135,136].

Abnormalities in cerebello-cortical circuits have been identified in mood disorders, with evidence suggesting that cerebellar dysfunction contributes to mood instability, impulsivity, and cognitive slowing [134,137]. Notably, in BD, cerebellar alterations have been associated with difficulties in regulating emotional responses and executive dysfunction, potentially worsening mental symptoms during manic and depressive episodes. These findings indicate that the cerebellum plays a broader role in neuropsychiatric disorders beyond motor coordination, reinforcing the need for further research into cerebellar-targeted interventions for mood disorders.

### 2.7. Corpus Callosum and Interhemispheric Communication

The corpus callosum, the brain’s largest white matter tract, is responsible for interhemispheric communication between cortical regions. Studies in MDD and BD reveal reduced corpus callosum integrity, which correlates with cognitive dysfunction, emotional dysregulation, and impaired social cognition [138,139]. Diffusion tensor imaging (DTI) studies suggest that white matter abnormalities in the corpus callosum disrupt connectivity between hemispheres, which may underlie deficits in attention, working memory, and emotional processing commonly seen in mood disorders [140,141].

Significant changes in corpus callosum morphology have been observed in individuals with OCD, possibly contributing to rigid cognitive processing and impaired executive function [142,143]. Additionally, studies indicate that reductions in corpus callosum white matter integrity are associated with depressive symptom severity, suggesting that interhemispheric dysconnectivity is central to mood regulation [141,144]. These findings highlight the importance of white matter pathways in mood disorders and suggest that targeting corpus callosum connectivity deficits could be a promising area for future interventions.

Table 2 summarizes the structural and functional brain changes observed in mood disorders, detailing key regions implicated in emotional regulation, cognitive processing, and interhemispheric communication. The table provides an overview of the neuroanatomical alterations, associated mood disorder symptoms, imaging techniques used for their identification, and key references supporting these findings.

## 3. The Neuronal Atrophy Hypothesis: Structural and Functional Impacts of Neuronal Atrophy in Mood Disorders

### 3.1. Structural Manifestations in Neurons

The neuronal atrophy hypothesis provides a unique perspective regarding structural manifestations within the brain that accompany mood disorders. This idea posits that chronic stress, genetic predisposition, and other environmental factors can cumulatively instigate a series of cellular changes, leading to shrinkage of neurons and disruption of neural communication networks [86,92,145]. This process reduces overall gray matter volume, intricate dendrite morphology, and synaptic connectivity changes [146,147,148] (Figure 1).

Additionally, neuroimaging reveals widespread alterations in white matter integrity in BD that are associated with deficits in executive function, processing speed, verbal fluency, and emotional regulation [149,150,151]. Some studies have demonstrated that individuals with depression may exhibit decreased dendritic arborization and density of synapses in brain regions, including the prefrontal cortex, a critical executive function structure [88,152]. Within the prefrontal cortex, diminished connectivity and dendritic complexity have been shown to impair its ability to modulate emotional responses and cognitive functions such as decision-making and problem-solving [153]. Additionally, altered functional integration across the prefrontal cortex and limbic structures may produce rigid negative biases, ultimately causing increased processing of negative information and decreased processing of positive information [107,154]. Other studies have revealed atrophy in the hippocampus, a brain region implicated in memory consolidation and emotional processing, in those with mood disorders, underscoring the neuronal atrophy hypothesis as a plausible mechanism underlying the pathophysiology of mood disorders [145,155,156,157]. Moreover, increased reactivity to emotional stimuli causes alterations in the amygdala, which may heighten emotional sensitivity and responsiveness to stressors, amplifying mood symptoms and rendering individuals more susceptible to mood episodes [158]. An inverse relationship between amygdala volume and the number of depressive episodes also occurs [159]. Table 2 presents a summary of these structural changes in the brain.

### 3.2. Intersecting Neurobiological Models in Mood Disorders

Neuronal atrophy intersects with other neurobiological models of mood disorders, elucidating the intricate interplay between different pathways and mechanisms underlying these complex conditions. For instance, the monoamine hypothesis posits that dysregulation in neurotransmitters like serotonin and norepinephrine contributes to the pathophysiology of mood disorders [160]. Post-mortem studies have revealed low serotonin levels in the brains of depressed patients in comparison to non-depressed patients [161]. In addition, many existing antidepressant therapies such as serotonin reuptake inhibitors (SSRIs), serotonin and norepinephrine reuptake inhibitors (SNRIs), tricyclic antidepressants (TCAs), and noradrenergic and specific serotonergic antidepressants (NaSSAs) have been developed based on the monoamine hypothesis [162]. Disrupting the integrity of neural circuits involved in synthesizing, releasing, and reuptake of monoamines could exacerbate neuronal atrophy dysregulation in monoamine neurotransmitters, thereby perpetuating mood symptoms. Monoamine neurotransmitters and most antidepressants have proven to work through astrocytes [163], as summarized in comparisons across neurobiological models (Table 1). Astrocytes are involved in many brain functions in both active and passive roles, and they are the most abundant cells in the brain [164,165,166]. Growing evidence demonstrates astrocyte loss in key limbic system regions in depressed patients [163,165].

Additionally, the neuroinflammatory hypothesis implicates immune dysregulation in the onset and progression of mood disorders, with chronic inflammation potentially exacerbating neuronal atrophy by releasing pro-inflammatory cytokines and oxidative stress. Patients with BD have demonstrated central and peripheral elevations in pro-inflammatory elements, including acute-phase reactants, cytokines, chemokines, prostaglandins, and oxidative/nitrosative species, in addition to increased expression of inflammatory genes and aberrant complement and cellular activation [167,168,169,170]. By considering these interconnected pathways, clinicians and researchers can develop more comprehensive treatment approaches targeting neurotransmitter imbalances and structural and inflammatory changes within the brain, ultimately improving outcomes for mood disorders. Figure 1 depicts an overview of these ideas.

## 4. Molecular Mechanisms of Neuronal Atrophy in Mood Disorders

### 4.1. Neurotrophic Factors and Signaling Pathways

The role of BDNF and its receptor TrkB is critical to synaptic plasticity, neuronal survival, and structural integrity. BDNF binding to TrkB activates multiple signaling pathways, including PI3K/Akt/mTOR, PLCγ, and Ras/ERK, all contributing to synaptic remodeling and neuronal resilience [171,172,173]. The BDNF/TrkB/PKC pathway is essential for neurotransmission and synaptic maintenance [174]. Furthermore, the PI3K/Akt/mTOR pathway is critical in dendritic branching and synaptic plasticity, key factors in preventing neuronal atrophy [102].

Long-term potentiation (LTP) is vital for learning and memory, and it depends on BDNF signaling [175]. In mood disorders, dysregulated BDNF-TrkB signaling is observed, contributing to synaptic dysfunction and neuronal atrophy [176]. Severely depressed patients exhibit low levels of BDNF in the prefrontal cortex and hippocampus, correlating with hippocampal atrophy, neuronal apoptosis, and synaptic loss [177,178]. Antidepressants have been shown to increase BDNF levels and enhance neurogenesis, supporting the neurotrophic hypothesis of depression [179,180,181].

### 4.2. HPA Axis Variability and Cortisol Dysregulation

Chronic stress exposure alters hypothalamic–pituitary–adrenal (HPA) axis activity, leading to cortisol dysregulation. However, cortisol responses in mood disorders are heterogeneous: some patients exhibit HPA hyperactivity, while others display blunted cortisol responses [182,183]. These variations suggest a complex regulatory system involving glucocorticoid receptor sensitivity (*NR3C1* mutations) and epigenetic modifications in *FKBP5*, which modulate stress adaptation [184,185].

Sustained elevated cortisol levels have been linked to hippocampal dendritic atrophy, while chronic-stress-induced reductions in cortisol impair neurogenesis and synaptic remodeling [186,187,188,189,190,191]. In rodent models, prolonged glucocorticoid exposure reduces *MAP2* and spinophilin expression, markers of dendritic spine integrity [192]. In humans, hippocampal volume reduction in depression correlates with HPA dysfunction, highlighting the neurotoxic effects of prolonged stress exposure [193].

### 4.3. IL-33/ST2 Pathway in Neuroinflammation and Mood Disorders

Neuroinflammation is increasingly recognized as a driver of neuronal atrophy in mood disorders. The IL-33/ST2 pathway, primarily expressed in astrocytes and oligodendrocytes, regulates microglial activation and cytokine release [194,195,196,197,198]. IL-33 deficiency is associated with enhanced microglial activation, synaptic loss, and increased vulnerability to stress-induced depressive behaviors [195,198]. In contrast, IL-33 overexpression attenuates neuroinflammation and protects against neuronal atrophy [196,197].

Additionally, the ST2 receptor, which mediates IL-33 signaling, is downregulated in the prefrontal cortex of MDD patients, implicating the IL-33/ST2 axis as a potential biomarker for inflammatory-driven depression [199]. Targeting this pathway through immunomodulatory therapies presents a novel avenue for treating neuroinflammation-associated mood disorders [200].

### 4.4. Microglial Activation and Synaptic Pruning in Mood Disorders

Microglia, the resident immune cells of the central nervous system, play a crucial role in maintaining synaptic integrity and neuronal health [201,202]. Under homeostatic conditions, microglia facilitate synaptic remodeling and neuroprotection [203,204]. However, chronic stress and neuroinflammation induce a shift toward a pro-inflammatory phenotype, leading to excessive synaptic pruning and neuronal atrophy [99,100].

Activated microglia secrete inflammatory cytokines such as IL-6, TNF-α, and IL-1β, contributing to dendritic spine loss and impair neuroplasticity [205]. Elevated microglial reactivity has been observed in the prefrontal cortex and hippocampus of patients with major depressive disorder, suggesting a direct link between microglial activation and structural brain changes [100,206]. Additionally, microglial phagocytosis of synapses is heightened in stress-related mood disorders, exacerbating cognitive and emotional dysfunction [207].

Potential therapeutic interventions targeting microglial dysfunction include minocycline, a tetracycline derivative with anti-inflammatory properties, and fractalkine signaling modulators, which help regulate microglial activity and prevent excessive synaptic pruning [208]. Further research is needed to elucidate the precise molecular pathways through which microglia contribute to mood disorder pathology and to develop targeted therapies to mitigate their harmful effects.

### 4.5. Mitochondrial Dysfunction and Synaptic Integrity

Mitochondria are essential for neuronal metabolism, ATP synthesis, and synaptic maintenance. Dysfunction in mitochondrial oxidative phosphorylation and electron transport chain (ETC) complexes leads to oxidative stress, impaired neuroplasticity, and neuronal apoptosis [209]. Recent proteomic and metabolomic studies provide insights into specific mitochondrial abnormalities associated with mood disorders. Plasma neuronal extracellular vesicle (EV) analyses from MDD patients reveal decreased NRF2, CYPD, and MFN2 levels, all essential for mitochondrial function [210]. Metabolomic profiling further highlights dysregulated citric acid cycle metabolites, such as α-ketoglutarate and succinate, indicating impaired neuronal energy metabolism [211].

Post-mortem analyses of mood disorder patients have identified mitochondrial ETC activity dysfunction, particularly in Complex I and Complex IV. Reduced electron transfer efficiency in these complexes leads to ATP depletion, increasing neuronal vulnerability to metabolic stress [212]. The inability to sustain synaptic ATP demands results in impaired neurotransmitter release, synaptic vesicle cycling, and long-term potentiation (LTP), ultimately contributing to synaptic failure and atrophy [213].

Chronic stress induces mitochondrial fragmentation and impairs mitophagy, leading to excessive reactive oxygen species (ROS) accumulation and subsequent neuronal atrophy [214,215]. ROS accumulation triggers oxidative damage to mitochondrial DNA (mtDNA), proteins, and lipids, further exacerbating mitochondrial dysfunction [216]. This process disrupts calcium homeostasis, leading to mitochondrial permeability transition pore (mPTP) opening, cytochrome c release, and caspase activation, initiating apoptosis in vulnerable neuronal populations [217]. In parallel, stress-induced impairment of mitophagy—regulated by the PINK1/Parkin pathway—prevents the clearance of damaged mitochondria, compounding cellular toxicity and synaptic dysfunction [218].

The neuroprotective effects of coenzyme Q10, nicotinamide riboside, and mitochondrial-targeted antioxidants suggest promising therapeutic avenues for treating mitochondrial dysfunction in mood disorders [219,220,221,222,223,224,225,226]. These compounds enhance mitochondrial biogenesis, restore electron transport efficiency, and mitigate oxidative stress [219]. Preclinical studies indicate that targeting mitochondrial metabolism can prevent dendritic atrophy, increase synaptic spine density, and restore synaptic transmission in stress-exposed cortical and hippocampal neurons [227].

Furthermore, the role of *MFN2* in mitochondrial fusion is critical for maintaining synaptic energy balance [228]. *MFN2* dysfunction leads to excessive mitochondrial fragmentation, which reduces the availability of functionally competent mitochondria at synapses [229]. This results in synaptic instability and impaired neuroplasticity, hallmark features of mood disorders [215]. Inhibition of mitochondrial fission or enhancement of mitochondrial fusion have been proposed as a potential neuroprotective strategy to sustain synaptic integrity under chronic stress conditions [230].

The binding of BDNF to TrkB activates downstream intracellular pathways, including:PI3K/Akt/mTOR pathway: facilitates neuronal survival and synaptic plasticity by promoting dendritic branching and spine formation.PLCγ pathway activates calcium-dependent mechanisms, including ER Ca^2+^ release and CaMKII/CaMKIV. These mechanisms phosphorylate CREB to regulate gene expression, promoting synaptic remodeling and long-term potentiation (LTP).The Ras/Raf/ERK/MEK pathway regulates the gene activity necessary for synaptic maintenance, long-term potentiation, learning, and memory.

## 5. Integrative Models

Although several studies have focused on individual aspects of the neuronal atrophy hypothesis, the molecular mechanisms discussed collectively contribute to this model [231]. The distinct physiological mechanisms form a complex network of structural interactions within the brain that produce the phenotypic mood disorder. For example, chronic stress and elevated glucocorticoid levels not only impact neuronal integrity but can also influence the expression of inflammatory cytokines and disrupt mitochondrial function—two other mechanisms discussed that contribute to neuronal atrophy [232,233]. Chronic inflammation can exacerbate mitochondrial dysfunction and further harm neurotrophic signaling pathways [234,235] (Table 3). These alterations in neurotrophic signaling can also influence glucocorticoid sensitivity, which modulates the stress response and increases neuroinflammation and destruction [236,237]. This intersection between these individual factors resulting in neuronal atrophy reflects the multifaceted nature of mood disorder pathophysiology. It emphasizes the need for a more comprehensive analysis of the mechanisms involved.

Genetic and environmental factors also play critical roles in modulating these molecular mechanisms and the neuronal atrophy hypothesis [238,239,240,241]. The genetic predisposition of individuals may influence the expression of molecules involved in neurotrophic signaling, the stress response, and other pathways involved in neuronal atrophy. For example, polymorphisms in genes encoding neurotrophic factors such as BDNF increase susceptibility to mood disorders and alterations in neuroplasticity [175,237,242,243,244]. Furthermore, genetic variations in glucocorticoid receptor sensitivity influence individual stress responses and vulnerability to mood disorders [245,246,247,248]. The protein FKBP5 is associated with gene mutations and epigenetic changes that result in different phenotypic expressions of psychiatric disease. It has even been proposed as a target for new therapeutic agents [246,248]. Environmental components of neuronal atrophy have also been analyzed, including the relationship between early-life trauma and lifestyle factors and epigenetic modifications that alter gene expression and contribute to mood disorder pathogenesis [249,250]. Understanding this interplay between these components will be essential in elucidating the pathophysiology of mood disorders and developing targeted therapeutic interventions.

Additionally, resistance training contributes to neurotrophic support, though its effects on BDNF are modest compared to aerobic exercise [251]. Combining aerobic and resistance exercise may yield synergistic effects, enhancing neuroplasticity and synaptic maintenance [252]. Beyond neurotrophic signaling, exercise also mitigates the deleterious effects of chronic stress and inflammation by regulating glucocorticoid levels and reducing pro-inflammatory cytokines like IL-6 and TNF-α [251,252]. Incorporating lifestyle interventions, particularly regular physical activity, may be an effective strategy for restoring BDNF levels, counteracting stress-induced neuronal atrophy, and mitigating the inflammatory and mitochondrial dysfunction pathways contributing to mood disorder pathophysiology [251,252,253,254].

## 6. Therapeutic Implications

### 6.1. Conventional Pharmacological Treatments for Mood Disorders

The therapeutic implications of the neuronal atrophy hypothesis for mood disorders are evident. By further investigating the different molecular mechanisms of mood disorders, new pharmacologic agents may be created that target mood disorder treatment in novel ways. The neuronal atrophy hypothesis posits that in targeting the primary excitatory and inhibitory pathways of neuronal activation, a faster-acting and more effective antidepressant treatment can be developed, one that would surpass the slower onset and modest effect of selective serotonin reuptake inhibitors (SSRIs), the current mainstay of mood disorder treatment [255]. The mechanistic underpinnings of SSRI treatment for mood disorders are an increase in serotonin in the synaptic cleft, allowing for the improvement in mood disorder symptoms [256]. In recent years, however, the justification has moved away from this previous line of thinking—the current prevailing school of thought is that SSRIs play a role in a complex downstream effect, ultimately resulting in changes in neural plasticity at the structural and physiological level [257]. Mechanisms of how SSRIs can induce such neural changes include alterations in the expression of critical genes such as CREB and BDNF that lead to changes in the activation of 5-HT receptor subtypes [255,256,257]. Additionally, recent findings by Rantamäki and Castrén suggest that antidepressants directly bind to the transmembrane region of the TrkB receptor, acting as allosteric modulators that enhance BDNF signaling and promote neuronal plasticity, further expanding our understanding of SSRI mechanisms [258]. However, attempts to further elucidate and distinguish a clear molecular pathway for SSRI function have yielded limited success [257]. Additionally, SSRIs are notorious for their required lengthy trial period before seeing therapeutic effects, as well as the litany of side effects associated with medication use, discontinuation, and abrupt cessation [256].

Emerging evidence highlights exercise as a critical protective factor that counteracts these mechanisms and promotes BDNF release. Aerobic exercise and high-intensity interval training (HIIT) significantly enhance BDNF levels [179,259] through activity-dependent pathways involving calcium influx and downstream signaling cascades, such as PI3K/Akt/mTOR, and ERK/MEK [254,260]. For example, aerobic exercise improves hippocampal neurogenesis, synaptic plasticity, and cognitive resilience [254,260]. HIIT, in particular, produces rapid, acute increases in circulating BDNF levels compared to lower-intensity modalities, further emphasizing its efficiency [253] (see Figure 2).

Targeting the glutamate and GABA receptor pathways of neuronal activity is a possible therapeutic benefit. Specifically, preclinical studies show that agents targeting glutamate pathways, such as glutamate-positive allosteric modulators (PAMs) and glutamate receptor antagonists, have a more rapid onset of antidepressant effect in rodent models [261,262,263]. Unfortunately, clinical trials to reproduce these results in the patient setting are limited and notably unsuccessful due to multiple reasons, one being that glutamate PAMs can have more side effects, and finding the balance between improved therapeutic effects and limited adverse effects has been challenging [261].

### 6.2. Alternative Pharmacological Approaches for Treatment-Resistant Depression (TRD)

Ketamine, an NMDA channel blocker, is emerging as a possible avenue of more effective mood disorder treatment. Ketamine shows a dose-dependent relationship with glutamate release, in which rapid glutamate bursts are observed with ketamine use [264]. This ketamine-dependent glutamate burst also interacts with signaling pathways such as mTORC1 and BDNF, causing increased synapse formation [264,265,266]. Ketamine shows remarkable results as an anxiolytic for severe, treatment-resistant generalized anxiety disorder, with a dose-dependent relationship between ketamine and anxiolytic effects [264,266,267]. As such, these unconventional treatments could soon prove to be the next frontier of MDD treatment. Another promising area of exploration involves the use of classic psychedelics, such as psilocybin and the entactogen/empathogen MDMA, which shows potential in promoting neuronal plasticity and improving mood disorders [268,269]. Classic psychedelics induce structural and functional changes in the brain, enhancing synaptogenesis and neuritogenesis [72,73,270]. Ultimately, promoting neuritogenesis, the growth of new neural connections, represents the most promising treatment approach, offering a potential pathway to reversing the neuronal atrophy observed in mood disorders.

While selective serotonin reuptake inhibitors (SSRIs) and serotonin-norepinephrine reuptake inhibitors (SNRIs) remain the mainstay of pharmacological treatment for mood disorders, older classes of antidepressants, such as tricyclic antidepressants (TCAs) and monoamine oxidase inhibitors (MAOIs), continue to play a role in treatment-resistant depression (TRD) [271,272,273,274,275,276,277]. TCAs, including amitriptyline, nortriptyline, and clomipramine, block the reuptake of serotonin and norepinephrine while also affecting histaminergic and cholinergic receptors. This broad pharmacological activity contributes to their strong antidepressant efficacy but also results in significant side effects, including cardiotoxicity, sedation, and weight gain, which limit their widespread use [278,279,280,281,282]. Similarly, MAOIs, such as phenelzine and tranylcypromine, inhibit monoamine metabolism, thereby increasing neurotransmitter levels [283,284]. However, these drugs require strict dietary restrictions due to their interaction with tyramine-containing foods, which can lead to hypertensive crises [285,286]. Despite these drawbacks, TCAs remain a viable option for patients with comorbid chronic pain or insomnia, while MAOIs are particularly useful for atypical depression characterized by hypersomnia and weight gain. However, both are generally reserved for treatment-resistant cases due to their significant side effect profile [287,288].

### 6.3. Anti-Inflammatory Agents: A Novel Approach to Mood Disorders

Emerging evidence suggests that targeting neuroinflammation may provide a novel avenue for mood disorder treatment. Increased levels of pro-inflammatory cytokines such as IL-6, TNF-α, and IL-1β often characterize chronic inflammation, which has been implicated in neuronal atrophy and synaptic dysfunction in depression [99,289]. Several pharmacological agents with anti-inflammatory properties show potential as adjunctive treatments for depression. Non-steroidal anti-inflammatory drugs (NSAIDs), particularly celecoxib (a selective COX-2 inhibitor), have demonstrated antidepressant effects when used alongside SSRIs [290,291]. Additionally, biologic agents targeting cytokines, such as TNF-α inhibitors (e.g., etanercept) and IL-6 antagonists (e.g., tocilizumab), have exhibited promising effects in individuals with elevated inflammatory markers. However, while these agents show potential, their routine clinical use in depression remains investigational, warranting further large-scale trials to assess long-term efficacy and safety [292,293]. Another emerging candidate is minocycline, a tetracycline antibiotic with anti-inflammatory properties that reduces microglial activation and glutamate excitotoxicity, mechanisms implicated in mood disorders and neurodegeneration [294,295]. These findings suggest that anti-inflammatory therapies may be particularly beneficial for a subset of depressed patients with inflammation-driven pathophysiology.

### 6.4. Mitochondrial Dysfunction and Metabolic Approaches to Depression

Another promising avenue involves mitochondrial-targeted interventions, given increasing evidence linking mitochondrial dysfunction to depression. Neurons are highly dependent on mitochondrial ATP production, and deficits in mitochondrial metabolism have been observed in patients with major depressive disorder (MDD), particularly in the prefrontal cortex and hippocampus [296,297]. Coenzyme Q10 (CoQ10), a key electron carrier in the mitochondrial respiratory chain, reduces oxidative stress and improves energy metabolism in depression models [298,299]. Similarly, creatine, which serves as a cellular energy buffer, has efficacy in SSRI-resistant depression, possibly by enhancing ATP production and synaptic function [300,301]. Another promising compound, acetyl-L-carnitine (ALCAR), appears to regulate the epigenetic modifications of BDNF, a neurotrophic factor implicated in synaptic plasticity and neuroprotection [302,303]. Given that metabolic dysfunction and chronic fatigue frequently co-occur with depression, mitochondrial modulators could serve as adjunctive treatments to enhance treatment efficacy and mitigate residual symptoms in treatment-resistant patients. These therapeutic methods are summarized in Table 4.

### 6.5. Future Directions: Biomarker-Guided Personalized Psychiatry

As research continues to explore novel pathways beyond the traditional monoaminergic model of depression, integrating approaches that target inflammation, metabolism, and synaptic plasticity may lead to more effective and personalized treatments for mood disorders. These findings highlight the need for further clinical trials to refine the role of anti-inflammatory and mitochondrial-targeting therapies in psychiatric care. Future research should focus on identifying reliable biomarkers that predict the response to these interventions, facilitating a more personalized approach to depression treatment and optimizing therapeutic outcomes.

## 7. Challenges and Limitations

The complexity of mood disorders presents limitations in the neuronal atrophy hypothesis. Mood disorders involve multiple interacting pathways, including stress response, circadian rhythms, and synaptic function [304,305,306]. Genetic studies reveal two distinct transdiagnostic liabilities differentiating common psychopathologies from serious mental illnesses, leading to difficulty in treatment strategies [307]. Neuroimaging research shows both shared and distinct neural mechanisms in bipolar and unipolar disorders [308]. Environmental factors, such as climate change and urbanization, significantly impact brain health and mental illness [309]. Additionally, this hypothesis struggles to account for the heterogeneity of mood disorders and variety of structural brain changes seen in individuals with these conditions. This hypothesis also focuses heavily on structural loss but may underestimate the role of functional neuroplasticity, synaptic remodeling and neurotransmitter dynamics that may be equally important in mood regulation. Furthermore, there is no direct and widely accepted clinical biomarker for neuronal atrophy in mood disorders, which leads to difficulty in evaluating and validating this hypothesis in live human subjects.

Further, the methodological challenges in neuropsychiatric research emphasize the need for more robust approaches. Cross-sectional designs and small sample sizes limit the ability to capture dynamic relationships between neuronal atrophy and mood disorders [310,311]. Longitudinal and integrative approaches are crucial for understanding these complex processes [310,311]. Large sample sizes are necessary to detect small effects and ensure reliability in brain–behavior correlations [312]. Neurobiological differences between depressed individuals and healthy controls are often small and have limited predictive utility [313]. Dimensional and data-driven approaches may improve sensitivity in identifying brain–behavior associations [314]. Collaborative efforts like the DIRECT consortium aim to address reproducibility issues by pooling large datasets and standardizing analysis pipelines [315]. Future trials should focus on longitudinal designs and modifiable targets to inform prevention and treatment strategies [316].

## 8. Methodological Framework for Longitudinal and Integrative Research

Understanding neuronal atrophy in mood disorders requires a comprehensive research approach integrating longitudinal neuroimaging, molecular profiling, genetic analysis, and computational modeling. Investigating how chronic stress, inflammation, and neurotrophic signaling affect neuronal structure over time will allow for the identification of biomarkers, risk factors, and therapeutic targets. A combination of advanced imaging techniques, biomarker analyses, and experimental models is necessary to distinguish whether neuronal atrophy is a primary cause of mood disorders or a consequence of chronic dysregulation in affected brain regions. Establishing multimodal research protocols will enable the development of targeted interventions that prevent or reverse neuronal atrophy in affected populations [317,318].

### 8.1. Longitudinal Neuroimaging Studies

Tracking neuronal atrophy requires longitudinal neuroimaging studies that assess structural and functional changes across different stages of illness. High-resolution structural MRI enables volumetric analysis of the prefrontal cortex, hippocampus, amygdala, thalamus, and basal ganglia, regions critically involved in mood regulation [319,320]. Diffusion tensor imaging (DTI) provides insights into white matter integrity and structural connectivity, helping to determine whether neuronal atrophy affects communication between key neural circuits [321]. Resting-state and task-based functional MRI (fMRI) should be used to evaluate functional network disruptions in corticolimbic circuits that regulate emotion and cognition [322,323]. Magnetization transfer imaging (MTI) and myelin water imaging (MWI) can be employed to assess myelin degradation and neuroinflammatory processes that contribute to structural atrophy [324,325].

In addition to MRI-based methods, positron emission tomography (PET) should be used to measure glucose metabolism (FDG-PET) and neuroinflammation (TSPO-PET), providing molecular-level insights into cellular dysfunction [326,327]. These imaging techniques should be applied at multiple time points (e.g., baseline, 6 months, 12 months, and beyond) to determine the trajectory of neuronal atrophy and assess the impact of pharmacological and non-pharmacological interventions [328,329]. Using large-scale, multi-cohort imaging datasets, such as those available from the ENIGMA consortium, will facilitate reproducibility and cross-population comparisons [330].

### 8.2. Biomarkers of Neuronal Atrophy

Molecular biomarkers complement neuroimaging findings by identifying the biological mechanisms driving neuronal atrophy. Measuring the serum and cerebrospinal fluid (CSF) levels of BDNF and TrkB receptor expression can provide insights into neuroplasticity and synaptic integrity [331,332]. Cortisol and *FKBP5* methylation levels indicate hypothalamic–pituitary–adrenal (HPA) axis dysregulation, a key factor in stress-induced neuronal atrophy [189,333]. Cytokine profiling should include markers such as interleukin-6 (IL-6), tumor necrosis factor-alpha (TNF-α), and C-reactive protein (CRP) to assess the contribution of chronic neuroinflammation to neuronal degeneration [334,335]. Evaluating oxidative stress markers, such as malondialdehyde, glutathione, and superoxide dismutase, will help determine the extent of mitochondrial dysfunction and excitotoxicity in affected individuals [336,337].

Multi-omic approaches should integrate proteomic, metabolomic, and lipidomic profiling to establish more precise biomarker signatures associated with mood disorders. Single-cell RNA sequencing (scRNA-seq) can help characterize transcriptional changes in neurons and glial cells, revealing cell-type-specific vulnerability to stress-induced atrophy. These biomarkers should be evaluated in prospective, large-scale studies, enabling the development of personalized treatment strategies based on an individual’s molecular profile [338,339].

### 8.3. Genetic and Epigenetic Contributions

Genetic and epigenetic factors influence susceptibility to stress-induced neuronal atrophy and modulate the brain’s ability to recover from structural damage. Genome-wide association studies (GWASs) have identified risk alleles linked to *BDNF*, *FKBP5*, *NR3C1*, and other neuroplasticity-related genes, suggesting that genetic predisposition is critical in neuronal resilience. Transcriptomic and proteomic analyses using RNA sequencing and mass spectrometry can reveal dysregulated pathways in synaptic plasticity and neuronal metabolism [340,341].

Epigenetic modifications, including DNA methylation, histone acetylation, and non-coding RNA regulation, should be examined to understand how early-life stress and environmental exposures contribute to mood disorder susceptibility. Longitudinal epigenetic studies will help determine whether stress-induced DNA modifications are reversible and whether interventions such as antidepressants, exercise, or dietary changes can restore regular gene expression profiles. Integrating multi-omic datasets across diverse patient populations will be essential for identifying biologically relevant subtypes of mood disorders and stratifying patients based on their risk of developing severe neuronal atrophy [342,343].

### 8.4. Experimental Models for Mechanistic Validation

Experimental models are essential for validating causal mechanisms underlying neuronal atrophy in mood disorders. Rodent models of chronic stress, such as chronic unpredictable stress (CUS) and social defeat stress (SDS), provide insights into dendritic complexity, synaptic loss, and behavioral correlates of mood dysregulation. Imaging-based approaches, such as two-photon microscopy and in vivo calcium imaging, can track real-time synaptic changes in response to stress exposure.

Human-derived induced pluripotent stem cells (iPSCs) and brain organoids allow for the study of neuronal and glial interactions in stress-related pathology. CRISPR-Cas9 genome editing can manipulate *BDNF*, *FKBP5* and *NR3C1* to determine their direct effects on neuronal survival and plasticity. Incorporating in vitro electrophysiological techniques will provide functional insights into how stress alters neuronal excitability and synaptic transmission. These findings should be correlated with human neuroimaging and biomarker data to ensure translational relevance [344,345].

### 8.5. Computational and Predictive Modeling

Computational approaches can help synthesize multi-modal data from neuroimaging, molecular, and genetic studies to develop predictive models of neuronal atrophy progression. Machine learning algorithms trained on large-scale datasets can identify patterns in structural connectivity, molecular changes, and clinical symptoms that predict disease severity and treatment response.

Artificial-intelligence-driven network neuroscience approaches should be employed to model functional and structural connectivity disruptions in mood disorders [346]. Multivariate predictive modeling can stratify patients based on their risk of neuronal atrophy and likelihood of responding to specific interventions [347]. These computational tools will help optimize precision medicine approaches, allowing clinicians to tailor treatment strategies based on individual neurobiological profiles [348].

### 8.6. Translating Research into Precision Medicine

Future clinical trials should incorporate multimodal biomarker panels and neuroimaging endpoints to track treatment-induced reversal of neuronal atrophy and ensure clinical relevance. Stratifying patients based on their molecular and neuroimaging profiles will improve the efficacy of pharmacological, neuromodulatory, and behavioral interventions. Neuroplasticity-enhancing treatments, such as ketamine, psilocybin, and transcranial magnetic stimulation (TMS), should be investigated for their ability to restore dendritic complexity and synaptic function [349,350,351,352,353]. Anti-inflammatory therapies targeting cytokine dysregulation may benefit individuals with elevated neuroinflammatory markers. Integrating these approaches into personalized treatment protocols will help mitigate the impact of neuronal atrophy on mood disorder progression [354,355].

Future research incorporating longitudinal neuroimaging, molecular profiling, genetic analysis, and computational modeling will improve diagnostic precision, enable early intervention, and advance neuroprotective treatment strategies. Expanding this understanding will facilitate the development of biomarker-driven, mechanistically informed approaches for managing mood disorders and mitigating their global burden [356,357].

## 9. Conclusions

The neuronal atrophy hypothesis provides a critical framework for understanding the structural and functional alterations underlying mood disorders. By integrating findings from neuroimaging, molecular research, and genetic studies, this review highlights the role of neurotrophic factor dysregulation, neuroinflammation, mitochondrial dysfunction, and synaptic impairments in mood disorder pathogenesis. Advances in biomarker identification, targeted pharmacological interventions, and neuromodulatory therapies offer promising avenues for improving treatment outcomes. Future research should focus on longitudinal, multimodal investigations to refine diagnostic tools, personalize treatment strategies, and develop neuroprotective interventions that mitigate the long-term impact of mood disorders.

## Figures and Tables

**Figure 1 ijms-26-03219-f001:**
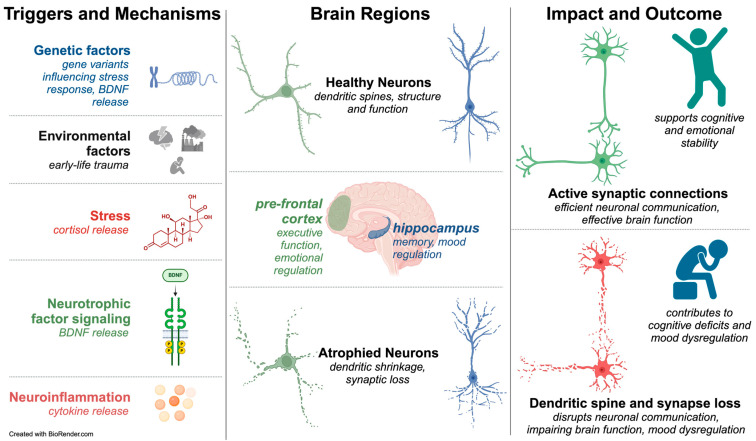
From stress to synapse: molecular and structural drivers of neuronal atrophy in mood disorders. Triggers and mechanisms (**left**): *Genetic factors*: variants in genes regulating stress response (e.g., *FKBP5*) and neurotrophic signaling (e.g., *BDNF*) increase vulnerability to stress and neuronal damage. *Environmental factors*: early-life trauma and chronic stress can initiate molecular and cellular changes that predispose neurons to atrophy. *Stress* (*cortisol release*): chronic stress activates the HPA axis, leading to prolonged cortisol exposure, negatively impacting neuronal health. *Neurotrophic factor signaling* (*BDNF release*): reduced levels of BDNF impair neuroplasticity, synaptic maintenance, and dendritic complexity. *Neuroinflammation* (*cytokine release*): elevated inflammatory cytokines like IL-6 and TNF-α exacerbate neuronal dysfunction and synaptic loss. Brain regions (**middle**): The prefrontal cortex (associated with executive function and emotional regulation) and the hippocampus (involved in memory and mood regulation) are highly vulnerable to these triggers, showing structural and functional changes during chronic stress or inflammation. Healthy neurons maintain intact dendritic spines, enabling proper synaptic function and communication. Atrophied neurons exhibit dendritic shrinkage and synaptic loss, which impair neuronal connectivity. Impact and outcome (**right**): Active synaptic connections in healthy neurons support efficient neuronal communication, contributing to cognitive and emotional stability. In contrast, loss of dendritic spines and synapses disrupts neuronal communication, impairing brain function and contributing to cognitive deficits and mood dysregulation.

**Figure 2 ijms-26-03219-f002:**
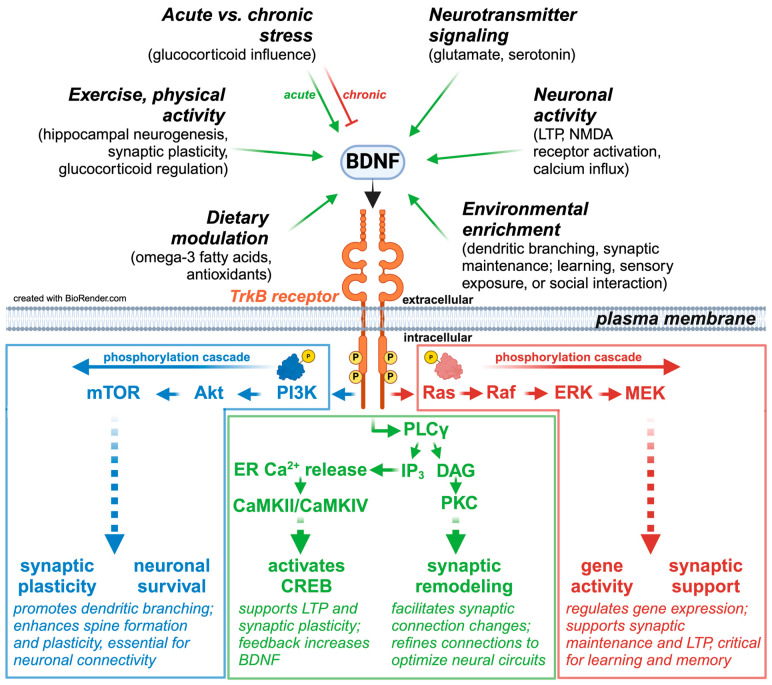
Molecular pathways and modulation of brain-derived neurotrophic factor (BDNF) in neuronal function and mood regulation. This model illustrates the role of BDNF and its receptor, TrkB, in modulating synaptic plasticity, neuronal survival, and gene expression, emphasizing its involvement in mood disorders and neuronal atrophy. Various factors influence BDNF release and signaling, including acute and chronic stress (glucocorticoid influence), exercise (which enhances hippocampal neurogenesis and synaptic plasticity), dietary modulation (e.g., omega-3 fatty acids and antioxidants), neurotransmitter signaling (glutamate and serotonin), neuronal activity (e.g., NMDA receptor activation), and environmental enrichment (e.g., sensory exposure and social interaction).

**Table 1 ijms-26-03219-t001:** Key neurobiological models in mood disorders.

Model	Main Hypothesis	Major Molecular and Cellular Mediators	Criticism/Limitations
Monoamine Hypothesis	Mood disorders are due to neurotransmitter dysregulation (serotonin, norepinephrine)	5-HT receptors, norepinephrine, dopamine	Does not account for structural changes; slow response to SSRIs
Neuroinflammatory Hypothesis	Chronic inflammation disrupts neurogenesis, contributing to mood disorders	IL-6, TNF-α, microglia, astrocytes	Inconsistent findings across populations
Mitochondrial Dysfunction Hypothesis	Impaired mitochondrial function leads to neuronal energy deficits, excitotoxicity	Oxidative phosphorylation, ETC, ROS	Limited clinical trials; need for biomarkers
Neuronal Atrophy Hypothesis	Mood disorders arise from synaptic loss and structural changes due to stress	BDNF, glucocorticoids, cytokines	Requires more longitudinal evidence in humans

References are found in Section 1.2, Section 1.3 and Section 1.4.

**Table 2 ijms-26-03219-t002:** Neurostructural changes in mood disorders.

Brain Region	Observed Structural Change	Associated Mood Disorder Symptoms	Imaging/Analysis Technique
Prefrontal Cortex	Reduced dendritic arborization and synaptic density	Impaired decision-making, emotional dysregulation	MRI, post-mortem analysis
Hippocampus	Atrophy, decreased neurogenesis	Memory impairments, cognitive deficits, emotional imbalance	MRI, volumetric studies
Amygdala	Altered volume (often reduced)	Heightened emotional sensitivity, increased reactivity to stressors	MRI, PET
Thalamus	Altered functional connectivity, decreased volume	Altered sensory and emotional integration	Functional MRI, connectivity analysis
Basal Ganglia	Reduced striatal volume, impaired reward processing	Dysregulated motivation and reward processing	MRI, functional connectivity studies
Orbitofrontal Cortex	Reduced activity, impaired decision-making	Deficits in impulse control and affect regulation	fMRI, PET
Cerebellum	Decreased volume, motor and cognitive dysfunction	Coordination and cognitive control deficits	MRI, fMRI
Anterior Cingulate Cortex	Structural thinning, reduced functional connectivity	Disrupted emotional self-regulation	MRI, structural analysis
Corpus Callosum	White matter integrity loss, reduced interhemispheric connectivity	Impaired interhemispheric communication	DTI, MRI
White Matter	Reduced integrity	Deficits in executive function, processing speed, emotional regulation	Diffusion Tensor Imaging (DTI)

References are found in Section 2.1, Section 2.2, Section 2.3, Section 2.4, Section 2.5, Section 2.6 and Section 2.7.

**Table 3 ijms-26-03219-t003:** Molecular mechanisms involved in neuronal atrophy and mood disorders.

Mechanism	Molecular Pathway	Impact on Neuronal Atrophy	Implications for Mood Disorders
Brain-Derived Neurotrophic Factor (BDNF)	BDNF-TrkB Signaling, PI3K/Akt/mTOR, MAPK/ERK	BDNF deficiency reduces synaptic plasticity, dendritic branching, and neurogenesis	Low BDNF linked to depression and treatment-resistant mood disorders; antidepressants increase BDNF
Glucocorticoids and HPA Axis Dysregulation	HPA Axis Activation, Cortisol Signaling	Chronic cortisol exposure shrinks dendrites, impairs neurogenesis, and increases excitotoxicity	HPA hyperactivity found in major depression; cortisol elevation correlates with hippocampal shrinkage
Neuroinflammation (Cytokines and Microglia)	IL-6, TNF-⍺, Microglia Activation, IL-33/ST2	Pro-inflammatory cytokines impair neurogenesis, increase apoptosis, and contribute to microglial overactivation	Inflammation correlates with depressive symptoms; anti-inflammatory treatments show antidepressant potential
Mitochondrial Dysfunction	Electron Transport Chain, Oxidative Stress, Mitophagy	Impaired mitochondrial function disrupts ATP production, increases ROS, leading to synaptic damage and neurodegeneration	Mitochondrial dysfunction affects neuronal energy metabolism in bipolar disorder and major depression

References are found in Section 4.1, Section 4.2, Section 4.3, Section 4.4 and Section 4.5.

**Table 4 ijms-26-03219-t004:** Pharmacological and neuromodulatory treatments for mood disorders—mechanisms, efficacy, and limitations.

Treatment	Mechanism of Action	Efficacy	Side Effects	Limitations
Selective Serotonin Reuptake Inhibitors (SSRIs)	Inhibits serotonin reuptake to increase serotonin levels	Effective for mild to moderate depression; limited for TRD	Nausea, sexual dysfunction, emotional blunting	Delayed onset; high relapse rates in TRD
Serotonin-Norepinephrine Reuptake Inhibitors (SNRIs)	Inhibits serotonin and norepinephrine reuptake	Similar to SSRIs but may benefit some TRD cases	Increased blood pressure, withdrawal effects	Similar to SSRIs; withdrawal effects can be severe
Tricyclic Antidepressants (TCAs)	Blocks serotonin and norepinephrine transporters	Effective but high side effect profile limits use	Cardiotoxicity, sedation, weight gain	High toxicity; not first-line due to side effects
Monoamine Oxidase Inhibitors (MAOIs)	Inhibits monoamine oxidase enzyme, increasing neurotransmitter levels	Used in treatment-resistant cases; significant dietary restrictions	Hypertensive crisis (with certain foods), insomnia	Significant dietary restrictions; hypertensive risk
Ketamine (NMDA Antagonist)	Modulates glutamatergic neurotransmission via NMDA receptor blockade	Rapid-acting for TRD; effects last days to weeks	Dissociation, potential for abuse	Expensive; requires specialized administration
Psychedelics (Psilocybin, LSD)	Agonizes 5-HT2A receptors, promoting neuroplasticity	Potential for long-term remission; ongoing clinical trials	Hallucinations, altered perception, potential for misuse	Legal restrictions; long-term effects unclear
Transcranial Magnetic Stimulation (TMS)	Uses magnetic fields to stimulate brain regions involved in mood regulation	Moderately effective for TRD; non-invasive	Mild headaches, scalp discomfort	Requires multiple sessions; inconsistent results
Deep Brain Stimulation (DBS)	Electrically stimulates targeted deep brain structures	Potential in severe TRD; requires surgical implantation	Surgical risks, brain hemorrhage (rare)	Invasive procedure; ethical concerns
Anti-inflammatory Agents	Reduces neuroinflammation via cytokine modulation	Emerging data suggest benefits, but requires further validation	Potential immune suppression, unknown long-term risks	Lack of large-scale trials; variable responses
Mitochondrial Modulators (CoQ10, Creatine)	Enhances mitochondrial function and ATP production	Promising adjunctive therapy; limited clinical trial data	GI distress, unknown long-term effects	More studies needed on efficacy and dosing

References are found in Section 6.1, Section 6.2, Section 6.3, Section 6.4 and Section 6.5.

## Data Availability

No new data were created or analyzed in this study. Data sharing is not applicable to this article.

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
