# Peer review of "From Stress to Synapse: The Neuronal Atrophy Pathway to Mood Dysregulation"

_ijms, 2025, doi:10.3390/ijms26073219_

Round 1

Reviewer 1 Report

Comments and Suggestions for Authors

Central hypothesis of this review article is that mood disorders like major depressive disorder and bipolar disorder may arise from neuronal atrophy caused by chronic stress and related neurobiological changes, impairing brain regions critical for mood regulation. The review explores mechanisms such as dysregulated neurotrophic factors (e.g., BDNF), stress-induced glucocorticoid elevation, neuroinflammation, and mitochondrial dysfunction, which collectively disrupt synaptic plasticity and exacerbate mood dysregulation. It highlights the need for integrative research and discusses potential therapeutic strategies, including neuroplasticity-enhancing drugs, lifestyle interventions, and anti-inflammatory treatments to improve outcomes and reduce the global burden of mood disorders.

While the topic of this review is of broad interest and the hypothesis strong, the article fails short in providing an in depth description of the biological underpinnings in support to this theory. In fact, the review remains mostly superficial without giving sufficient molecular details of the mechanisms underlying the pathological processes of neuronal atrophy in mood disorders. In addition, several concepts are repeated twice in the review, making it exceedingly long with respect to the actual information given. Thus, very major revisions are required before this article may be considered suitable for publication. Specifically:

Abstract. “The review advocates for longitudinal and integrative research...”. Actually, the sections advocating this approach (Par. 8, and Conclusions) are quite restricted and the description of the methodology is rather superficial. Thus, the review fails short in giving useful recommendations on the methodology needed for a longitudinal and integrative research in this field. It is suggested to either remove this claim from the abstract or to expand and implement Par.8.

Par. 1.1. (lines 40-44). Simply indicating the increase in incidence may be misleading if the actual overall incidence is missing. For instance, a 68% increase on 1% of the population affected, is very much different from an overall incidence increasing from 20% to 34% (i.e. 68% increase) of the adolescent population. Please, provide actual figures.

Par. 1.1. (line 45). “In the US, more than one in ten individuals...”. What age? Adults? Specify.

Par. 1.1. (lines 47). “Approximately 8 million individuals die per year...” Where? In the world? In one continent? Only in US? Specify.

Par 1.2/1.3/Par 1.4. In these paragraphs, the limitations of neurotransmitters dysregulation are discussed and thereafter, the neuronal atrophy hypothesis is introduced. What about the limitations in the mitochondrial and inflammation hypotheses? Two new paragraphs should be placed here. Also, the paragraphs require a better organization to make them more readable. It is suggested to reorganize the three paragraph on the neurotransmitters, mitochondria and inflammation hypotheses by first introducing the mainstream hypothesis, then explain its limitations, and finally, conclude with the author’s view. Additionally, the information regarding the involvements of other neurotransmitters beyond monoamines are insufficient. In particular, given that in Chapter 6 also glutamate and GABA are taken into consideration, the involvement of these two neurotransmitters in mood disorders requires a proper introduction. Add more information, here.

Par 1.4. Provide here, a definition of neuronal atrophy, right at the beginning of the paragraph.

Par. 2. The title sounds strange. Why “Background”? A better title could sound like: “Historical perspective on neuronal atrophy”. Or just use: “Structural brain changes in mood disorders”.

Chapter 2. (lines 151-163). The first paragraph from “The diagnosis of mood disorders...” to “more effective treatment in the future” appears misplaced. As it deals with very general considerations on mood disorders, it would be more logical to place it in the Introduction, right at Line 1 of Par.1.1. Then, the Par.2 would start with the sentence “The historical perspective of neuronal atrophy...” which would make much more sense.

Chapter 2. (lines 173-175). “Over time, ...mood disorders (92,93).” This sentence is a repetition of the first sentence of the subsequent paragraph and can be deleted.

Chapter 2. (lines 176-183). From “Contemporary neuroimaging...”. This part is rather superficial. If it is intended as just an introduction of subsequent paragraphs, then it is an unnecessary repetition. On the other hand, if it intends to provide information, then it misses its goal. More data are needed to answer to questions such as: Which molecular changes where historically found? What morphological changes were found in dendrites, spines, axons, somata? Are there regional or cell type specific changes? Are these morphological changes same or different in the different mood disorders? Are there age-related differences? The current literature provides detailed answers to these specific questions.

Par 3.1. It is illogical that this paragraph is not connected to Chapter 2. (lines 176-183), as apparently they deal with the same topic. Also this paragraph presents the same serious limitations of the previous one. Furthermore, reporting for all mood disoders only the anatomical data of prefrontal cortex, and hippocampus is a totally unacceptable oversimplication. In fact, it is well known among specialists of the field that the brain areas involved are in part different in the various disorders, such as  OCD (orbitofrontal cortex, ACC, striatum-caudate nucleus, thalamus and – in some patients – also amigdala) or MD (PFC, hippocampus, amigdala, ACC, thalamus, basal ganglia). In this respect, also Table 2 requires a thorough revision.

Par. 3.3. The whole paragraph appears as a repetition of concepts already presented in other paragraphs, while it does not provide any additional information, especially on the molecular mechanisms underlying the various models of mood disorders. Let alone the fact that it does not consider that the different mood disorders have differences, besides their similarities.

Chapter 4. I started reading this chapter with the expectation to be finally enlightened about the molecular mechanisms underlying neuronal atrophy in the different mood disorders. Each paragraph of this chapter (4.1, 4.2, 4.3 and 4.3) starts with a scholar description of the molecular biology of the mechanisms which are involved in mood disorders, with the attempt to emphasize their role in neuronal atrophy. However, evidence in support to the role of dysregulation of BDNF in neuronal atrophy is scarcely reported. There are instead, several very general statements with no real information, such as for instance:

In those with mood disorders, dysregulation of this BDNF-TrkB 287

signaling pathway has been observed in multiple studies, emphasizing the role of this 288

pathway in how neuronal atrophy and synaptic dysfunction lead to psychological dis- 289

turbance

 “Reduced levels of BDNF and alterations in TrkB receptor expression 292

have been implicated in the pathophysiology of mood disorders, highlighting the signifi- 293

cance of neurotrophin signaling in maintaining brain plasticity and emotional well-being. 294”

Furthermore, dysregulated neurotrophin signaling profoundly affects neuronal sur- 310

vival and synaptic plasticity in mood disorders [143]. .... 311

Dysfunction in neurotrophin signaling pathways compromises neuronal 316

viability, leading to structural alterations, including dendritic atrophy and synaptic loss. 317

 Moreover, aberrant neurotrophin signaling disrupts synaptic plasticity mechanisms 321

essential for learning and memory processes, exacerbating cognitive impairments com- 322

monly associated with mood disorders. ... 323

Since, the review gets into some detail regarding the different BDNF/TrkB signalling pathways and their different functional physiological effects, it would be logical here to deepen the analysis at the level of individual  neurotrophic signalling pathways which are actually compromised and the type of detrimental effects observed in the different mood disorders, in order to allow the reader to understand if the observed alterations are supporting - or not - the neuronal atrophy hypothesis.

Similarly, in Par. 4.2, Par 4.3 and Par 4.4 there is no description of the molecular mechanisms through which cortisol, inflammatory citokines and mitochondrial dysfunctions, respectively can cause neuronal atrophy, and reduce dendritic complexity and the number of dendritic spines, or affects neurogenesis. These paragraphs require a complete revision to overcome the current lack of in depth analysis.

Par. 4.3. The role of glial cells, in particular of astrocytes and microglia should discussed here, in the context of the role of inflammation in mood disorders.

 Par. 4.5. This paragraph does not sufficiently account the available literature on the defective neurogenesis in the different animal models of mood disorders.

Chapter 5. (line 504) The protein FKBP5 appears abruptly in this paragraph which is intended to integrate the different parts of this review. Any protein/molecular mechanisms presented for the first time in a paper should be adequately introduced. Why this protein has not been previously introduced and discusses in one of the previous paragraphs?  

Chapter 5. (lines 513-530). This paragraph is misplaced. It is suggested to move into the Chapter 6. “Therapeutic implications”.

Chapter 6. Studies by the laboratories of Rantamäki T. and  Castrén E. on the direct binding of antidepressants to the transmembrane region of TrkB receptor should be cited and commented as well.  

Chapter 6. What is the meaning of this sentence ?

Considering opposing theories of mood disorders' pathophysiology could yield 551

more productivity. 552

 Chapter 8. Unfortunately, the review does not provid useful recommendations on the methodology to be used for this longitudinal and integrative research to be applied in this field. As this section may represent a major added value of the whole review, it is suggested to to expand and implement this section.

Reviewer 2 Report

Comments and Suggestions for Authors

This review examines the neuronal atrophy hypothesis in mood disorders, suggesting that chronic stress causes both structural and functional changes in the brain. It discusses key mechanisms, including dysregulation of neurotrophic factors (such as BDNF), elevated glucocorticoids, neuroinflammation, and mitochondrial dysfunction, all of which impair synaptic plasticity and contribute to mood dysregulation. The review also addresses environmental factors, such as early-life stress and urbanization. It emphasizes the need for integrative research to refine disease models and proposes therapeutic strategies targeting neuroplasticity, such as novel pharmacological treatments, lifestyle interventions, and anti-inflammatory therapies, to enhance patient outcomes and better manage mood disorders.

A major criticism is the lack of mechanisms explaining the hypotheses of depression. The author mentions existing hypotheses that have already been explained in previous experimental studies and reviews.

For example, reports on the impact of stressors to the changes in HPA axis parameters are not sufficiently investigated. Clinical studies suggest that dysfunction of the HPA axis plays a key role in depression, with hyperactivity of this axis. However, some studies have also shown that plasma/serum glucocorticoid levels can be either reduced or unchanged in these patients. Which are the mechanisms of different deregulations of the HPA axis?

In addition to disrupted signaling via BDNF-TrkB proteins and the numerous signaling pathways they regulate, stress also impairs the interaction between TrkB and N-methyl-D-aspartate receptors (NMDAR). The specificity of interactions between BDNF-TrkB and NMDAR is associated with learning processes and cognitive functions. This aspect should be explained and included.

Regarding inflammatory cytokines, the IL-33/ST2 pathway is a part of the innate immune system and plays a role in regulating inflammation and immune responses. This aspect should be explained and included.

Regarding mitochondrial dysfunction, depression has been associated with disruptions in mitochondrial metabolite levels and energy metabolism within specific brain regions. Thus, please provide literature data on which mitochondrial metabolites are altered. Also, provide literature data from proteomic and metabolomic studies and mitochondrial dysfunction in depression.

Reviewer 3 Report

Comments and Suggestions for Authors

In this review manuscript, the authors present a comprehensive exploration of the neuronal atrophy hypothesis, which may provide a potential explanation for mood disorders. The review delves into various factors, including neurotrophic factors, inflammation, mitochondrial alterations, and disruptions in neurogenesis.

The manuscript is well-written and logically structured, presenting more information about an hypothesis regarding plasticity-related events associated with depression, bipolar disorder, and other conditions. However, I recommend the following revisions to enhance clarity, conciseness, and focus before publication:

The manuscript’s extensive length contains repetitive information, particularly in the general mechanisms or molecular pathways section, which could be better suited for a textbook. Additionally, some definitions, such as SES (line 47), are lacking.

Furthermore, the paragraph in lines 91-95 presents two independent ideas (TMS and DBS versus pharmacological treatments) that could benefit from clearer interconnection.

The effects of serotonin system activation on neuroplasticity should be further elucidated to provide a more comprehensive understanding of the undesirable consequences.

Lines 151-163 could be more appropriately integrated into the introduction section.

As an example of repetitive information, lines 176-184 include previously explained concepts, with only the amygdala’s role being novel. I suggest summarizing these sections and other potentially redundant information.

Section 3.3 exhibits a sudden shift in topic, which is more interrelated to the subsequent paragraph (astrocytes and inflammatory responses).

Section 4 incorporates information regarding general molecular pathways. This section could be summarized more concisely, as these pathways are common and may divert attention from the main theme. A table, such as Table 3, could effectively convey the mechanisms associated with the neuronal atrophy hypothesis to readers.

Section 7 lacks clarity regarding the limitations of the hypothesis. It primarily describes other factors that influence mood disorders rather than explicitly outlining the hypothesis’s constraints.

Round 2

Reviewer 2 Report

Comments and Suggestions for Authors

I am satisfied with the author's responses to my questions raised in my initial review.